# U-Net-Id, an Instance Segmentation Model for Building Extraction from Satellite Images—Case Study in the Joanópolis City, Brazil

**Fabien H. Wagner** [1,2,*], **Ricardo Dalagnol** [2], **Yuliya Tarabalka** [3,4], **Tassiana Y. F. Segantine** [1], **Rogério Thomé** [1] and **Mayumi C. M. Hirye** [5]

1 GeoProcessing Division, Foundation for Science, Technology and Space Applications—FUNCATE, São José dos Campos, SP 12210-131, Brazil; tassiyeda@funcate.org.br (T.Y.F.S.); consultor.geo@funcate.org.br (R.T.)
2 Remote Sensing Division, National Institute for Space Research—INPE, São José dos Campos, SP 12227-010, Brazil; ricardo.silva@inpe.br or ricds@hotmail.com
3 Luxcarta Technology, Parc d'Activité l'Argile, Lot 119b, 06370 Mouans Sartoux, France; ytarabalka@luxcarta.com
4 Inria Sophia Antipolis, CEDEX, 06902 Sophia Antipolis, France
5 Quapá Lab, Faculty of Architecture and Urbanism, University of São Paulo—USP, São Paulo, SP 05508-900, Brazil; ma.hirye@alumni.usp.br or mayhirye@hotmail.com
* Correspondence: wagner.h.fabien@gmail.com or fabien.wagner@funcate.org.br; Tel.: +55-(12)-3925-1387

**Abstract:** Currently, there exists a growing demand for individual building mapping in regions of rapid urban growth in less-developed countries. Most existing methods can segment buildings but cannot discriminate adjacent buildings. Here, we present a new convolutional neural network architecture (CNN) called U-net-id that performs building instance segmentation. The proposed network is trained with WorldView-3 satellite RGB images (0.3 m) and three different labeled masks. The first is the building mask; the second is the border mask, which is the border of the building segment with 4 pixels added outside and 3 pixels inside; and the third is the inner segment mask, which is the segment of the building diminished by 2 pixels. The architecture consists of three parallel paths, one for each mask, all starting with a U-net model. To accurately capture the overlap between the masks, all activation layers of the U-nets are copied and concatenated on each path and sent to two additional convolutional layers before the output activation layers. The method was tested with a dataset of 7563 manually delineated individual buildings of the city of Joanópolis-SP, Brazil. On this dataset, the semantic segmentation showed an overall accuracy of 97.67% and an F1-Score of 0.937 and the building individual instance segmentation showed good performance with a mean intersection over union (IoU) of 0.582 (median IoU = 0.694).

**Keywords:** instance segmentation; U-net; building detection; urban landscape

## 1. Introduction

In recent years, automatic extraction of cartographic features from very high resolution aerial and satellite images is undergoing a revolution thanks to advances brought by deep convolutional neural networks (CNN). The main advantage of these supervised CNNs is that they take raw data and automatically learn features through training with minimal prior knowledge about the task [1]. CNN accuracy is similar to human-level classification accuracy, but is consistent and fast, enabling rapid application over very large areas and/or through time [2]. These CNNs could support the fast

acquisition of accurate spatial information about city buildings, and subsequently the production of the building environment maps, which is of great important for urban planning and monitoring.

For this task of building extraction, two types of segmentation with CNN can be applied: semantic segmentation and instance segmentation. Semantic segmentation consists of attributing a class to each pixel of an image. This type of segmentation is reasonably well-resolved for building segmentation from very high resolution images. For example, for the Inria Aerial Image Labeling Dataset [3], which is a dataset of buildings for 10 cities with varied urban landscapes over two different continents and with two semantic classes, namely "building" and "not building", the most recent models show intersections over union (IoU) of 80.32% and an overall accuracy of 97.14% (https://project.inria.fr/aerialimagelabeling/leaderboard/). This is remarkable considering that the validation samples are constituted of cities not present in the training samples [3]. Among the deep learning frameworks for building semantic segmentation, the U-net architecture has proven to be a standard and has achieved the highest accuracy [4,5].

On the other hand, instance segmentation is currently the most challenging task for building extraction. Instance segmentation consists of identifying each instance of each object of interest in an image at the pixel level. For example, the classes are no longer "building" and "not building" but "not building", "building 1", "building 2", "building 3"... up to "building N", even if the buildings are adjacent. It is still a domain in development and, in recent years, several Deep Learning frameworks have been developed for instance segmentation, for example Mask R-CNN [6], Deep Watershed Transform [7], and DeepMask [8] (see [9] for an extensive review of recent improvements of these models). Currently, there are two principal methods for producing the instance segmentation. The first consists of seeking the region of each object by creating its bounding box and mask. For example, the Mask R-CNN algorithm, which is probably the most widely used method nowadays, consists of two distinct modules: the first module is Faster R-CNN [10], which attributes a label to the object and generates the bounding box to encapsulate the object, while the second module produces a mask of the object [6]. The most recent published works focusing on building instance segmentation have used Mask R-CNN and shown that it performs poorly for edge extraction and to preserve the integrity of the building instances [11,12]. The second method is based on the prediction of the the mask characteristics to further segment the instance in a posterior step. For example, the Deep Watershed Transform [7] creates an energy map of the image, where object instances are represented as energy basins. In post-processing, the different energy basins are separated and correspond to the object instances.

The deep learning models for instance segmentation have mostly been developed with images from large-scale online databases, such as IMAGENET [13] and PASCAL VOC [14]; from medical imagery datasets, such as images from the challenge at the International Symposium on Biomedical Imaging (ISBI); or from images obtained with cameras onboard cars, such as the Kitti Road Data [15]. However, very little research has been done for instance segmentation of remote sensing imagery. Remotely sensed images have a characteristic which is not commonly encountered in the other types of images and which complicates the application of region-of-interest based instance segmentation (e.g., Mask R-CNN): they can contain very large number of objects of interests. For example, a very high resolution satellite image of a city can easily contain thousands of houses and buildings. This characteristic complicates the use of iterative region proposal frameworks, such as Mask R-CNN, mainly because: (i) during processing, this model creates one activation layer per object, thus it can quickly surpass the computer capacity for images with high density of objects of interest [9,16]; and (ii) in remote sensing, images are not independent. For example, even if one image is clipped into several smaller tiles for prediction, the tiles will need to be merged, and deciding whether or not to merge the object bounding box overlapping the border is a subjective expert-based decision, which decreases the segmentation accuracy.

Here, we present a U-net based model for instance segmentation called U-net-id. This model is of the same type as the Deep Watershed Transform and segments the instances, even if they are adjacent, directly based on the characteristics of the mask. It is designed to be simple and efficient while resolving some of the limitations of the common instance segmentation frameworks. As it is

based entirely on the U-net model, the computational resources do not increase with the number of objects in the image, such as in a region-of-interest based model (e.g., Mask R-CNN). The U-net model is also known to be better (higher Dice score) than Mask R-CNN to predict the segmentation mask when applied on the same dataset [17]. Our model is also designed to remove all expert-based decisions, such as the number of objects, which has to be defined by the user in Mask R-CNN and influence results and the computational resources, and definition of a threshold value to separate instances such as in the Deep Watershed Transform [6,7]. In its first part, it consists of three separate U-nets for the object segment, the object border, and the inner segment of the object (the segment of the object reduced by a defined number of pixels). In its second part, it concatenates the three obtained activation layers of the U-nets, which go on three separated streams through two convolution layers and an activation layer to separately predict the segment, the border, and the inner segment. The individualization of the instance is made in post treatment by extracting the inner segments (which are unique and do not overlap) and adding a buffer to them corresponding to the number of pixels that exists between the segment and the inner segment (2 pixels). The U-net-id model presents two main novelties. First, the mask used for the segmentation consists of three bands that contain the segments, their inner segments (segment reduced by 2 pixels), and their borders. All these mask layers share 1 or 2 pixels of overlap. Second, the result of each U-net is combined with the two others by concatenation operation and followed by several convolution layers and returns an output which is an activation map of its mask layer. These two novelties add a lot redundancy to the mask and the model that seems to help the model to converge. That is, it not only learns the mask layer individually but also the spatial relation (overlapping) between the mask layers.

As a study case, the model was tested for instance segmentation of the buildings of Joanópolis-SP (Brazil) using a sample of manually labeled individual buildings and the corresponding very high resolution satellite image of WorldView-3 with 0.3 m of spatial resolution.

The model R code is available at https://doi.org/10.5281/zenodo.3716070.

## 2. Materials and Methods

### 2.1. Model Architecture

The inputs of the model are RGB images of 256 × 256 pixels and 0.3 m of spatial resolution, Figure 1. Each image goes through a data augmentation process (see Section 2.5.1) and is then cropped to the size of 160 × 160 pixels to serve as input to the three separate U-nets. The size of 256 × 256 was chosen for image rotation during the data augmentation to prevent any missing values in the 160 × 160 image. Then, the image goes to three different paths that each start with an independent U-net model (Figure 2). The three U-nets separately segment the object border, the object segment, and the inner segment of the object. The border and the inner segments are created directly from the object segment; that is, the border goes from 4 pixels outside the object segment to 3 pixels inside and the inner segment is the object segment reduced by 2 pixels. By doing this, the three masks present an overlap of 1–3 pixels. After this, the three activation layers of 160 × 160 pixels resulting from the U-nets are concatenated in each of the paths. The following steps are two convolutional layers with 64 and 32 filters, respectively. Finally, the predictions are made separately for the segment, the border, and the inner segment using the last convolutional layer with a sigmoid activation function. The instance individualizations are made in post treatment by extracting the inner segments (which are unique and do not touch each other) and adding to them a buffer of 2 pixels, that is, the number of pixels that exist between the segment and the inner segment (as they have been created). The model has a total of 25,994,598 parameters, of which 25,973,478 are trainable.

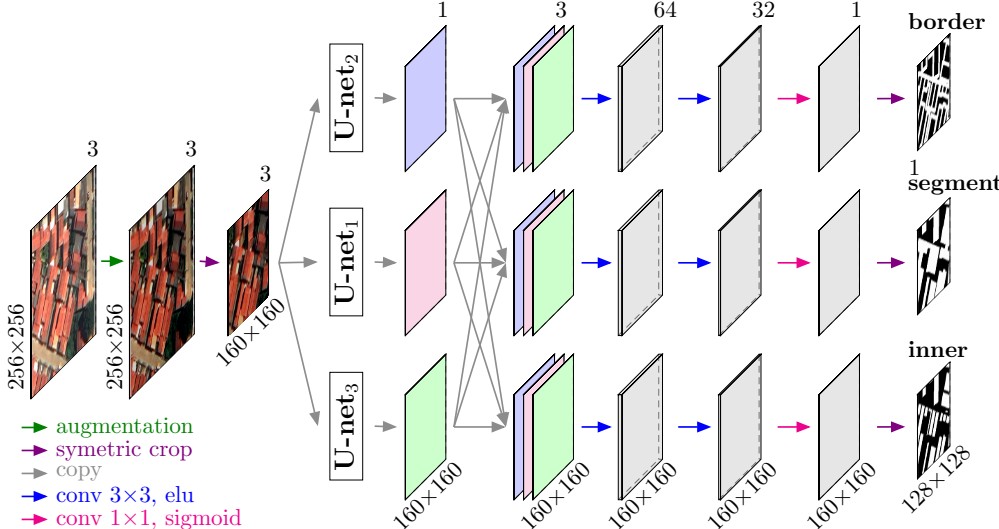

**Figure 1.** U-net-id model architecture. The three U-net units used in our model are identical but the three outputs of the U-net are represented with three different colors: magenta (U-net$_1$), blue (U-net$_2$), and green (U-net$_3$).

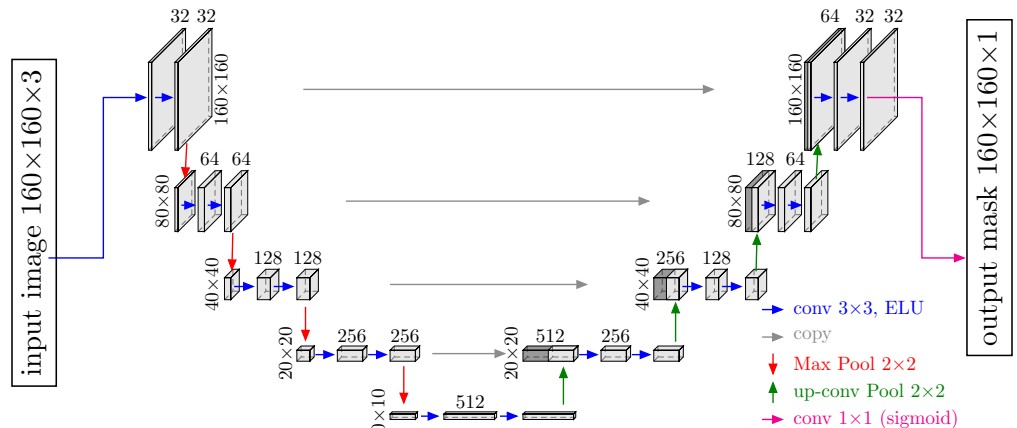

**Figure 2.** U-net model unit with characteristics of layer and filter sizes used in our model. The three U-net units used in our model are identical.

### 2.2. Study Site

The U-net-id was applied for instance segmentation of individual buildings in the city of Joanópolis-SP, Brazil. The city is a relatively low populated city for Brazil, with an estimated 2019 population of 13,220 inhabitants and a population density of 35.32 inhabitants per km$^2$ [18]. The city population has grown 12.34% between 2010 and 2019. The city's buildings are concentrated in an area of <25 km$^2$. The city's building landscape is mostly constituted of individual houses, a few tall buildings, and some large buildings for industrial, commercial, sport, or storage activities.

### 2.3. WorldView-3 Image

The WorldView-3 image (DigitalGlobe, Inc., USA) was acquired over the region on 4 July 2019, at an average off-nadir view angle of 19.8°. This image was distributed in one tile of 16,384 × 16,384 pixels that cover a region of ∼ 24.16 km$^2$. The spatial resolution was 0.3 m for the panchromatic band (464–801 nm) and 1.2 m for the selected multispectral bands: Red (629–689 nm), Green (511–581 nm), and Blue (447–508 nm). To prepare the image, all bands (multispectral and panchromatic) were first scaled between 0 and 2047 by their percentile 0.001 and 0.999, and then

scaled to 0–255 (8 bits). Second, the Red, Green, and Blue (RGB) bands were pan-sharpened with the panchromatic band using the Brovey algorithm [19] with a bilinear resampling (`gdal_pansharpen.py`) to create a single high-resolution RGB image with 0.3 m spatial resolution. No atmospheric correction was performed.

## 2.4. Building Dataset

To train the algorithm to detect and segment individual buildings, a mask was manually delineated using the WorldView-3 image of Joanópolis, SP, Brazil. The buildings were manually delineated, resulting in 7563 polygons that have all been validated in the field. Not all the buildings of the image were delineated; however, they were delineated entirely for most neighborhoods in the central region of the city. Each polygon represents one building. The images from the manual sampling presented different types of buildings: residential, commercial, and industrial. Using the delineated polygons, a raster mask coded in RGB (three bands) was produced (Figure 3) with the following values: background [0,0,0]; building segment [1,0,0] (Figure 3b); border segment [0,0,1] (Figure 3c), which is the border of the building segment augmented by 4 pixels outside and reduced by 3 pixels inside; and inner building segment [0,1,0] (Figure 3d), which is the building segment reduced by 2 pixels. The building inner segment is unique to each building and does not touch another inner segment (Figure 3d). The three masks present an overlap of at least 1 pixel. In the text, Layer Mask 1, Layer Mask 2, and Layer Mask 3 refer to building segments, border segments, and inner segments, respectively.

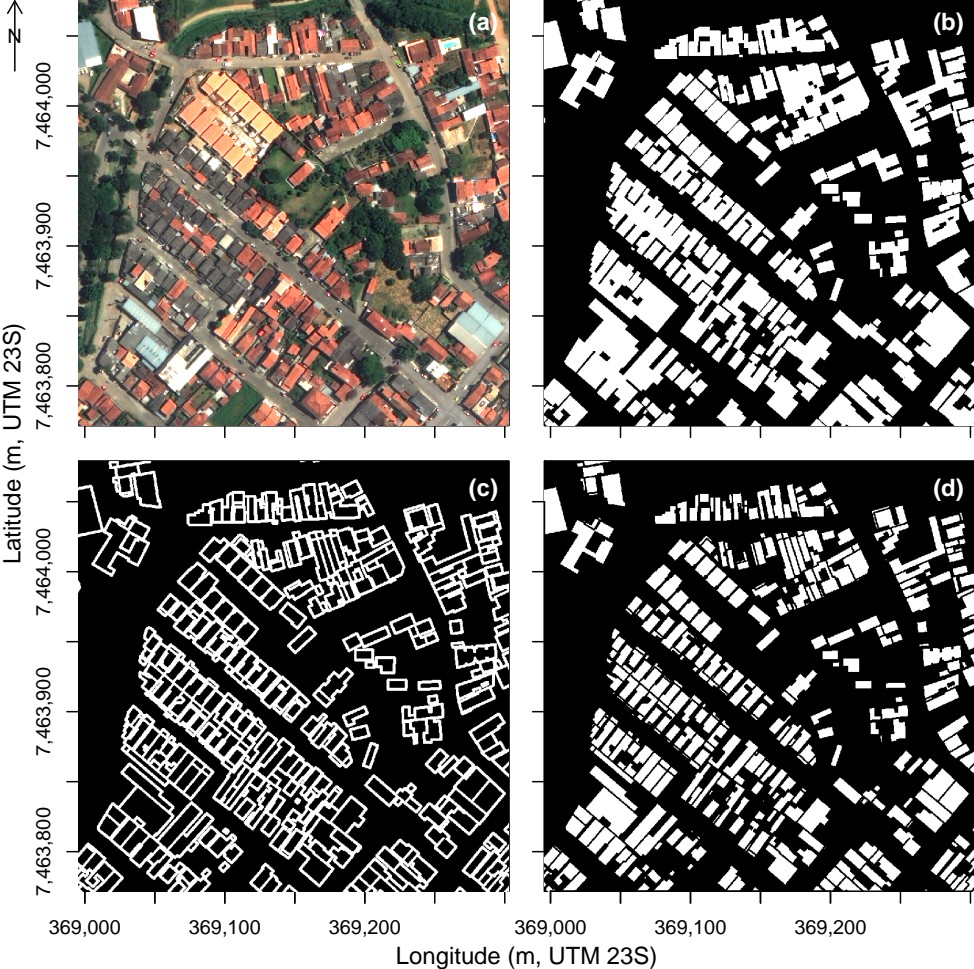

**Figure 3.** Example of a subset of the Joanópolis RGB image (**a**); and the corresponding masks for: building segments (**b**); borders (**c**); and inner segments (**d**). Satellite image courtesy of the DigitalGlobe Foundation.

## 2.5. Training

Clipping the image and the masks in 128 × 128 pixels over the region where buildings were manually delineated resulted in a sample of 2048 images and their associated labeled masks to train the model. Among these images, 1435 contained roofs and background and 613 contained only background. Then, 1638 images were used for the training and 410 for independent validation. The size of 128 × 128 pixels was selected because: (i) studied objects are generally smaller than 128 pixels in size (128 pixels = 38.4 m); (ii) the objects are not so dependant of a larger context; and (iii) we do not want the algorithm to learn a larger context. An example of large context would be 'houses always occur near asphalt streets'. The images were extracted from uniform grids of 128 × 128 pixels without any overlap between neighboring images. Then, 128 × 128 images were enlarged to 256 × 256 extents by adding 64 rows and columns on each side. Eighty percent of these images were used for training and 20% for validation. During network training, we used a standard stochastic gradient descent optimization. The loss function was designed as a sum of two terms: binary cross-entropy and Dice coefficient-related loss of the three predicted masks [20–22]. We used the optimizer RMSprop (unpublished, adaptive learning rate method proposed by Geoff Hinton here http://www.cs.toronto.edu/tijmen/csc321/slides/lecture_slides_lec6.pdf) with an initial learning rate of 0.01. We trained our network for 500 epochs, where each epoch comprised 68 batches with 12 images per batch.

### 2.5.1. Data Augmentation

Data augmentation was applied randomly to the input images, including 0–360° rotations, horizontal and vertical flips, and changes in the brightness, saturation and hue, by modulating the current values by 30–100% for brightness, 100–100% for saturation (no change), and 0–200% for hue.

### 2.5.2. Segmentation Accuracy Assessment

Three performance metrics were computed. First, the overall accuracy was computed as the percentage of correctly classified pixels. Second, the $F1$ score was computed for each class $i$ as the harmonic average of the precision and recall (Equation (1)), where precision is the ratio of the number of segments classified correctly as $i$ to the number of all segments (true and false positive) and recall is the ratio of the number of segments classified correctly as $i$ to the total number of segments belonging to class $i$ (true positive and false negative). This score varies between 0 (lowest value) and 1 (best value).

$$F1_i \;\; = \;\; 2 \times \frac{precision_i \times recall_i}{(precision_i + recall_i)} \tag{1}$$

Third, to estimate the accuracy of the instance segmentation, the intersection over union ($IoU$) metric was computed as the intersection of areas labeled as objects in the prediction and in the reference divided by the union of areas labeled as objects in the prediction and in the reference. To compute the $IoU$ of each object, we attributed to each individual object the predicted segment that showed the largest overlapping to the observed object.

### 2.5.3. Prediction

For prediction, the WorldView-3 (WV-3) tile of 16,384 × 16,384 pixels was clipped with regular grid with cells of 512 × 512 pixels and 64 neighbor pixels were added on each side to create an overlap between the patches. If there was a remaining blank portion (for example, due to the tile border), it was filled by the symmetrical image of the non-blank portion. The predictions were made on these images of 640 × 640 pixels, and the resulting images were clipped to 512 × 512 pixels and merged again to reconstitute the original 16,384 × 16,384 pixels WV-3 tile. This overlapping method was used to avoid border artifacts during prediction, a known problem for the U-net algorithm [5]. To belong to a given class, the pixel prediction value must be greater than or equal to 0.5. The instance segmentation mask was then produced by buffering the inner segment (mask Layer 2) by 2 pixels.

### 2.5.4. Algorithm

The model was coded in the programming language R [23] with Rstudio interface to Keras [21,22] and Tensorflow backend [24]. The training of the models took ∼2–20 h using GPU on an Nvidia RTX2080 with 8 GB of dedicated memory. Prediction using GPU of a single tile of 16,384 × 16,384 pixels (∼24.16 km$^2$) took approximately 10 min.

### 3. Results

For the city of Joanópolis, the algorithm presents a good level of segmentation accuracy (layer mask 1) with an overall accuracy of 97.67% and an F1-Score of 0.937 (precision = 0.936 and recall = 0.939) (Table 1). The detection rate on the validation data set was of 97.67%; that is, on the 2226 objects in the validation dataset, the algorithm found 2176 objects and missed 52. The mean intersection over union was 0.582 and the median was 0.694. Considering the entire WorldView-3 tile, the algorithm delineated and individualized 7477 buildings. The model segmentation for the three masks was very accurate, as seen by the segment in blue in Figure 4a–c. There are very few errors for the segment and inner segment, as seen in white in Figure 4a,c. The errors seem slightly higher for the border mask (Figure 4b). This error of the border is further propagated to the instance polygons (Figure 4d), but overall the instance segmentation can be considered as correct.

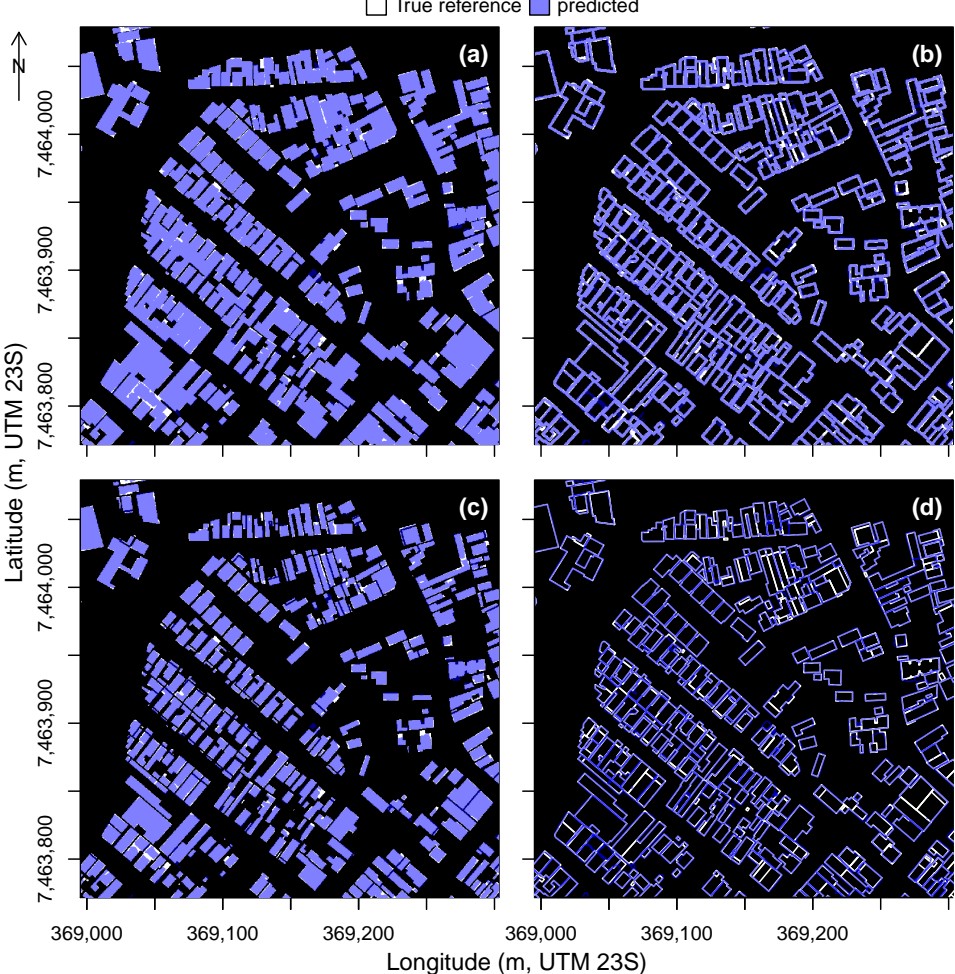

**Figure 4.** Example of the reference masks (in white) and predicted masks (in transparent blue) for: the segments (**a**); the borders (**b**); and the inner segments (**c**). The comparison between the reference building polygons and the predicted polygons generated by adding a buffer of 2 pixels around the polygon of the predicted inner mask is shown in (**d**).

**Table 1.** Segmentation accuracy of the Joanópolis buildings dataset.

| Overall Accuracy (%) | Precision | Recall | F1-Score | IoU Mean | IoU Median | Detection Rate (%) |
|---|---|---|---|---|---|---|
| 97.67 | 0.936 | 0.939 | 0.937 | 0.582 | 0.694 | 97.67 |

The model successfully learned the spatial relation of overlap between the mask layers, as shown in Figure 5. The overlap between the reference masks (Figure 5a) is very similar in the predicted mask (Figure 5b). In both figures, the outside border shows a size of 4 pixels (in blue), the inside border has a size of 3 pixels, and there are 2 pixels of overlap with only the segment mask (in pink) and 1 pixel of overlap with the inner segment mask and the segment mask (in white). Visually, the part with the most variation in the prediction is the outside part of the border (in blue) (Figure 5a). This is the only part which is not redundant in the other layers.

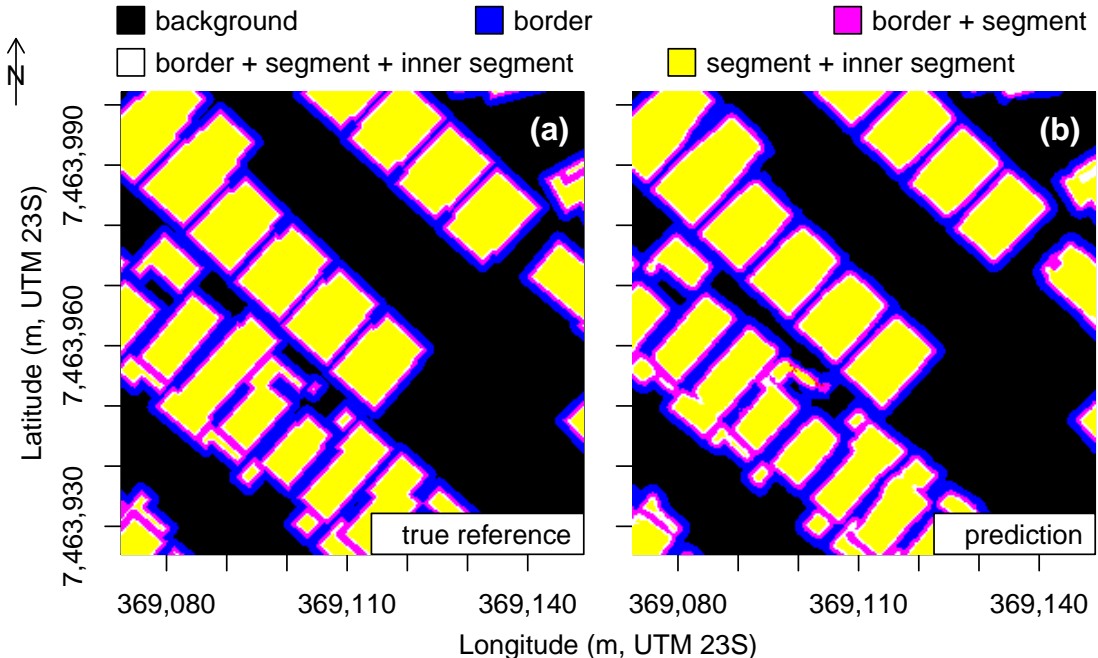

**Figure 5.** Example the reference masks and their overlap (**a**) and for the predicted masks resulting from the U-net-id model (**b**). The reference and prediction masks are RGB images with the following values: background [0,0,0]; building segment [1,0,0]; inner building segment [0,1,0], which is the building segment reduced by 2 pixels; and border segment [0,0,1], which is the border of the building segment augmented by 4 pixels outside and reduced by 3 pixels inside. All masks share a minimum of 1 pixel of overlap in the reference. The white color, which represents the overlap of 1 pixel among the segment, the border, and the inner segment, is found between the colors pink and yellow.

The polygons from the prediction appear slightly more rounded than the manually delineated polygons because the algorithm systematically misses the more extreme pixels at building corners (Figures 5b and 6). Some of the common errors are indicated by arrows in Figure 6b, a subset of the image that contains the main types of errors. Arrow 1 shows that a tree is above the roof and obscures the building features. In this case, the algorithm only detects the pixels presenting building features, as trees are considered to be background in the training sample. Arrow 2 shows a polygon that was delineated erroneously as a building by the producer and correctly ignored by the algorithm. Arrows 3–5 show some error of instance delineation, that is, where the model fails to separate the instances. The task of separating the instances can be extremely difficult (see Arrow 3): where the manual delineation was not aligned with the image features, the model found the building angle and started to make a border. However, likely due to the absence of characteristic building features,

it did not manage to separate the building into two instances. For Arrows 4 and 5, the roof features are very similar. Furthermore, looking at the manually delineated polygon in the bottom left corner, it can be observed that it shows contrasted building features in only one instance. The way the algorithm decides whether or not to segment these instances was likely based on the characteristics of the instances in the training sample.

In the main extension of Joanópolis (Figure 7 and Table 2), 6106 building instances are individualized (7380 buildings in the reference dataset). The buildings cover an area of 618,077 m$^2$. The mean area covered by a building in Joanópolis was found to be 101.2 m$^2$ (median 68.6 m$^2$) from the results of our model. For the same spatial extent, the manually delineated buildings area was of 86.6 m$^2$ (median 78.4 m$^2$).

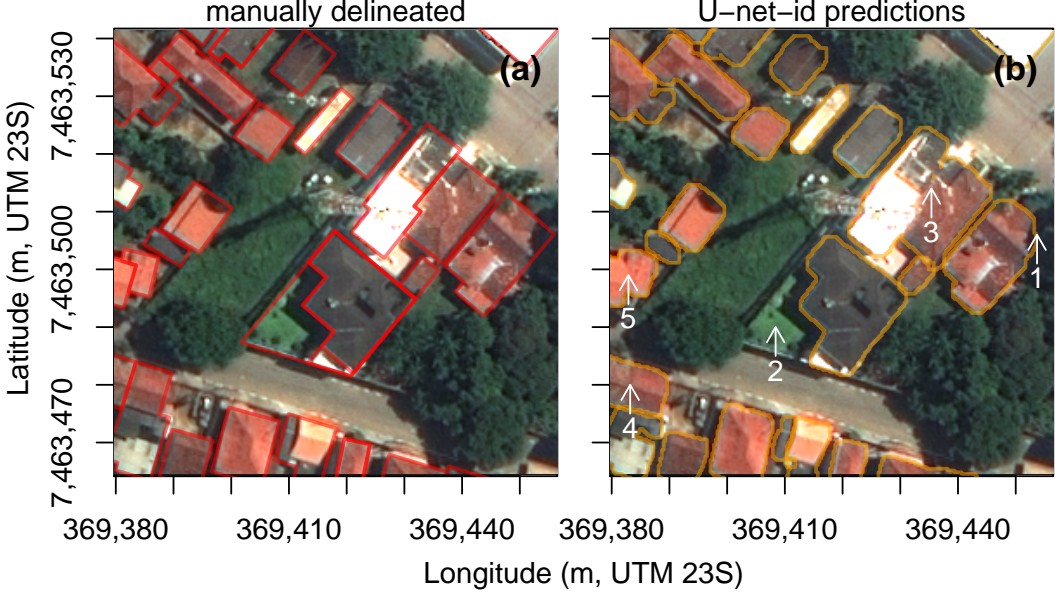

**Figure 6.** Example of a 256 × 256 Joanópolis RGB image with the manually delineated (**a**) and the predicted instance mask of buildings (**b**). This subset of the image has been selected because it contains several of the main errors of the algorithm or of the producer (highlighted by the white arrows). Satellite image courtesy of the DigitalGlobe Foundation.

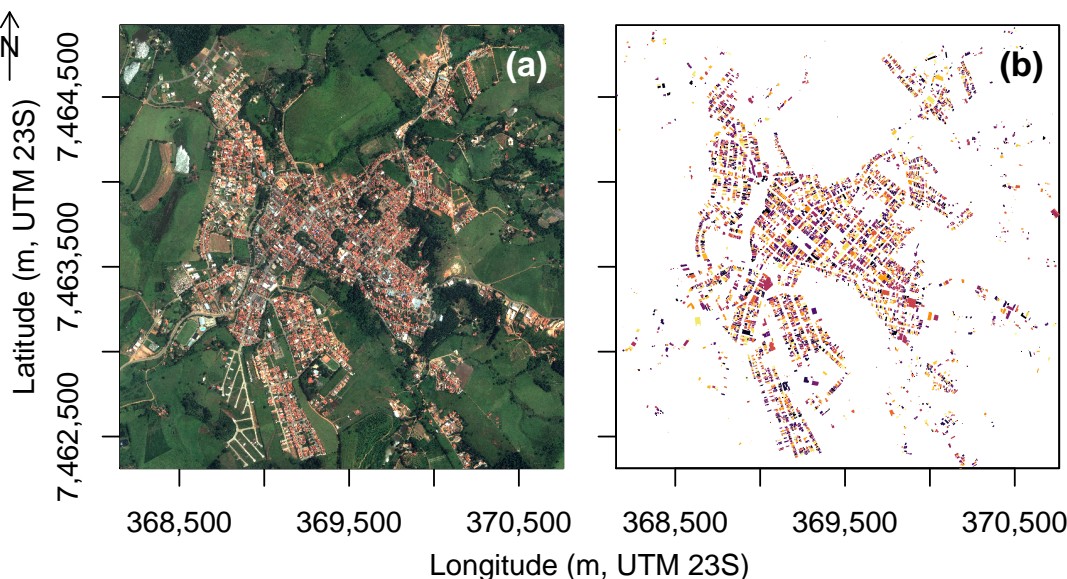

**Figure 7.** Main extension of Joanópolis (**a**) and the predicted instance mask of buildings (**b**). Satellite image courtesy of the DigitalGlobe Foundation.

**Table 2.** Comparison of the building numbers and areas between the Joanópolis building dataset (observed) and the buildings instances predicted by the U-net-id model (predicted).

|  | Observed | Predicted |
|---|---|---|
| *Worldview tile extent* | | |
| Number of buildings | unknown [a] but > 7563 | 7477 |
| *Joanópolis main extent* | | |
| Number of buildings | 7380 | 6106 |
| Building total area (m$^2$) | 639,285 | 618,077 |
| Mean building size (m$^2$) | 86.6 | 101.2 |
| Median building size (m$^2$) | 68.6 | 78.4 |

[a] In the reference dataset, not all buildings in the extent covered by the Worldview tile have been manually delineated.

## 4. Discussion

In this work, we propose a new instance segmentation neural network based on the U-net architecture called U-net-id (Figure 1). The main novelties of this model are, first, a structure with three parallel U-nets to predict the segment, the segment border, and the inner segment for each object. The border, the segment, and the inner segment of the object are all created from the mask of object instances with defined, constant overlaps between them. For example, the border overlaps with the segment and the inner segment are 3 and 1 pixel, respectively, and the inner segment is the segment reduced by 2 pixels. The mask used for the segmentation therefore comprises three bands, one for each different part. Second, the resulting activation layers of the U-nets, one for each path, are copied and concatenated (skip connection) on the three parallel paths, and a set of convolution layers is added that enables the network to learn the spatial overlap between the different parts of the object. That is, not only it has learned the mask layers individually but it has also learned the spatial relation of the overlap between the mask layers (Figure 5). Then, a sigmoid activation layer ended each path and separately predicted the segment, the segment border, and the inner segment. The two main characteristics of the model are the three correlated masks and the skip connections between the three paths after the U-net modules. These characteristics add redundancy that seems to help the model to converge and generalize. This is confirmed by the high performance of the model, with an overall accuracy of 97.67%, an F1-Score of 0.937 (precision = 0.936 and recall = 0.939), and a mean intersection over union of 0.582 (median IoU = 0.694). Furthermore, the median of area covered by a building in the prediction of 68.6 m$^2$ was close to the median of manually delineated building area of 78.4 m$^2$. Our model missed ∼17% of the instances (Table 2). This happened when two or more buildings were adjacent and not fully separated (see Figures 4d and 6b). However, as the objects are very small, the larger buildings are about ∼50 m (or 150 pixels) width, this value should decrease with object of increasing sizes. The Joanópolis building dataset is also complex; for example, it can have two instances under the same roof, which represents a reality in the field but a difficult or impossible case to resolve by the algorithm. Because the most recent works using Mask R-CNN to segment building instances are applied to different datasets [11,12], unfortunately, accuracies are not compatible with our results and comparison of the different methods on the same datasets will be made in a future work.

This model shares some similarities in the conception with the Deep Watershed Transform model [7], such as using the characteristics from the mask to find the instances and not relying on iterative strategies, such as in RNNs. Such as the Deep Watershed Transform model, our method does not handle objects that have been separated into multiple pieces by occlusion [7]. However, one advantage of our model over the Deep Watershed Transform model is that the object instances are already explicit in the masks, that is, each instance can be identified only with the inner segments. There is no need for non-Deep Learning post-processing steps to identify the building instances, which could lead to reduced accuracy; this can take place in the Deep Watershed Transform model [7]. Furthermore, as the model successfully manages to reproduce the overlap between the masks, the original segment can easily be produced by

adding a buffer of 2 pixels to the predicted inner mask individual polygons, which is exactly how the inner mask is created in the training sample. One other advantage is that all the instances (the inner segments) are simply coded in 0 and 1 in only one raster layer: there is no need for one raster mask per object or for storing the bounding boxes in vector data, such as in the Mask R-CNN [6]. In addition, as the inner segments are coded only in 0 and 1, adjacent images (sub-tiles of the large WorldView-3 tiles) can be merged without having to make a decision on how to merge the inner segments that are on the border. This is in contrast to merging two bounding boxes, for example, from Mask R-CNN, which will inevitably have different identifiers [6]. Furthermore, for Mask R-CNN, the number of objects to be found in the images must be given manually as a parameter (anchors) [6], and the results can differ based on this parameter. Our method does not have this limitation. It does not depend of a previous probability of object presence in the image, apply the same process to every pixel and is not sensitive to the number of buildings in the image. Finally, the binary masks have also some practical advantage since the loss function is only the sum between the binary cross entropy and the Dice coefficient related losses, computed with the predicted and the observed masks reshaped to a tensor of one dimension.

The model can suffer from classical problems that affect classification with multi-spectral satellite imagery, extremely dark shade (for example, a house totally in the shade of a building that is not visible in the image by human eyes), obstruction by cloud cover, or by trees (such as in Figure 6b). However, the model should not be sensitive to changes in atmospheric conditions and view-illumination effects (as long as the features of the buildings are visible) since the model is trained to give little importance to the buildings reflectance values, that is, in the data augmentation, the model is trained to recognize houses with a larger and varied range of hue, saturation, and brightness values than in reality. It was also observed that adding rotation in the data augmentation helped the model to generalize. Obviously, the Joanópolis dataset does not contain all the building morphologies that could be encountered around the world and the training sample has to be increased in order to improve generalization. However, the only information that users need to provide are additional images and mask to train the model, no parameters have to be adjusted by hand, such as in the currently most used instance segmentation frameworks such as Mask R-CNN or Deep Watershed Transform [6,25].

We found that the more frequent error of the algorithm was to systematically round the building angles (see Figures 5b and 6b). This error is recurrent in the CNN: for example, all the results of semantic segmentation in the leaderboard of the INRIA labeling dataset present this error, even those with the best accuracies (see https://project.inria.fr/aerialimagelabeling/leaderboard/). While this problem is not so important for the individualization of buildings and counting, it is critical for the use of such models for accurate surface measurements, as it systematically decreases the area of the buildings, and this problem gets worse as the size of object decreases. The explanation for this problem could be that while the CNN can reproduce a line, it is just an approximation of a curve to a line. It misses the conceptualization of the line as a human does, and consequently, this further complicates producing a sharp angle (an intersection of lines). As an analogy, it could be seen as inflating a balloon inside a cube, at some point, the balloon can take almost all the space and be perfectly flat where it is in contact with the cube sides. However, it will be difficult or even impossible to fit well inside the angles.

In future works, to solve this problem, two solutions could be tested: (i) giving more attention of the CNN on the angles, with additional mask(s), for example; or (ii) trying to make it better understand the concept of line, which could be made, for example, by enabling the model to see a larger image at each scale using larger CNN filters (more than $3 \times 3$ pixels) or $3 \times 3$ filters with stride. Note that this problem will likely occur predominantly for manmade objects but to a lesser extent for natural objects that mostly show smooth and round angles (e.g., tree crowns and rivers). Further research will be conducted to improve this. The other errors, such as the separation of the instances, could be likely corrected by adding more training samples and/or improving/correcting the existing training samples, without modifying the model. Future improvements will be made in these directions.

Individual buildings represent valuable data as they can be used to produce detailed urban cover maps and to characterize urban forms and to inform urban growth and sprawl. In addition,

individual building detection, alongside height data and usage data (whether residential or commercial, for example) are crucial to a finer spatial allocation of population, such as in a framework of gridded population data [26]. The model presented here is the result of a discussion with the Brazilian Institute of Geography and Statistics (IBGE), the institution which is in charge of the population census of Brazil (https://www.ibge.gov.br/en/highlights.html?destaque=26544). In future work, this methodology will be applied to map the individual buildings of the main Brazilian cities. In the case of the Brazilian population census, mapping individual buildings in areas of recent urban growth is required to inform the census operational plan, helping to define where and how many IBGE technicians must be sent to field to conduct the census (http://www.in.gov.br/web/dou/-/aviso-236116475). Due to the rapid urban growth of legal and illegal settlements in Brazilian cities, currently this could only be feasible in a reasonable time with automatic methods, such as the one developed here. As almost 90% of the world population growth is expected to occur in less-developed countries by 2100 [27], there is a need for fast and automatic mapping of human settlements and population estimates, and our model could be used for this purpose. In a future work, the method will be tested and trained on other Brazilian cities to better generalize to different urban patterns (e.g., low to high buildings density, slums and highly vegetated areas).

The model has been developed in R with the `keras` package [28,29] for RStudio [30], supporting that R can be used for Deep Learning model development. Our model is the first instance segmentation model made entirely from within the R environment. While for the strict Deep Learning part of the algorithm there is currently almost no difference between R and Python, that is, the R commands are translated and sent to Python with the help of the `keras` and `reticulate` packages [31], R possesses many convenient libraries for spatial data processing. The main spatial packages that have been used here are the `raster` package to work with the raster data [32], the `sf` package to work with spatial vector data [33], and tools to convert between vector data and raster data, such as the `fasterize` function [34] in R or outside R with function directly from the GDAL/OGR library [35]. The main time bottleneck with the spatial data processing remains the transformation of resulting raster of the model (16,384 $\times$ 16,384 at 0.3 m resolution) to vector data with `gdal_poligonize`, which can take around 20 min for one tile if there are many of objects (>50,000), even with a custom script for retiling the tile before the vectorization of the predicted objects. This conversion can be sped up with the function `isoband` from the R package `isobands` [36]; however, the border of the objects will be simplified during vectorization, resulting in some minor variation in the objects area between the raster and the shapefile (which is not the case when using `gdal_poligonize`). To conclude, R provides convenient and easy-to-use libraries for Deep Learning applications and development for remotely sensed imagery without the need to delve deeply into Python programming. On the other hand, due to the high similarity of the `keras` code in R and Python, a Python user will likely have no major difficulties converting the model from R to Python.

## 5. Conclusions

In this work, we present a new architecture of deep learning called U-net-id, specifically designed for the instance segmentation of buildings in very high resolution satellite imagery. The architecture is built to work with image in tiles, such as remote sensing images, and to reduce the minimum information that needs to be provided by the user: only the images for prediction or only the images and masks for training. No manual threshold has to be defined by the user, which is not the case in the currently most used instance segmentation frameworks such as Mask R-CNN or Deep Watershed Transform. The U-net-id architecture achieves great performance, for instance segmentation of the instance building dataset of the city of Joanópolis-SP, Brazil. It will be further improved with samples from other Brazilian cities and applied to building instance segmentation of the main cities of Brazil. The Joanópolis dataset does not contain all the building morphologies that can be encountered in Brazil or in other countries; however, to adapt the model to learn new features of buildings, the user will only need to provide additional training samples and the architecture will remain unchanged. Due to the model architecture simplicity, it could easily be reproduced and used for buildings or tested for

instance segmentation of other objects in very high resolution images. The model R code is available at
https://doi.org/10.5281/zenodo.3716070.

**Author Contributions:** Conceptualization, F.H.W., T.Y.F.S., R.T., and M.C.M.H.; methodology, F.H.W., R.D, Y.T., and M.C.M.H.; software, F.H.W., R.D., and R.T.; validation, F.H.W. and T.Y.F.S.; formal analysis, F.H.W.; investigation, F.H.W.; resources, F.H.W.; data curation, F.H.W., T.Y.F.S., and R.T.; writing—original draft preparation, F.H.W.; writing—review and editing, F.H.W., R.D, Y.T, T.Y.F.S., R.T., and M.C.M.H; visualization, F.H.W.; supervision, F.H.W.; project administration, F.H.W.; and funding acquisition, F.H.W. All authors have read and agreed to the published version of the manuscript.

**Funding:** The research leading to these results received funding from the project BIO-RED 'Biomes of Brazil— Resilience, Recovery, and Diversity', which is supported by the São Paulo Research Foundation (FAPESP, 2015/50484-0) and the U.K. Natural Environment Research Council (NERC, NE/N012542/1). F.H.W. has been funded by FAPESP (grant 2016/17652-9). R.D. acknowledges the support of FAPESP (grant 2015/22987-7). Y.T. has been funded by the project EPITOME ANR-17-CE23-0009 of the French National Research Agency (ANR). We also thank the Amazon Fund through the financial collaboration of the Brazilian Development Bank (BNDES) and the Foundation for Science, Technology and Space Applications (FUNCATE) no. 17.2.0536.1 (Environmental Monitoring of Brazilian Biomes).

**Acknowledgments:** We thank DigitalGlobe for the provision of WorldView-3 satellite images.

**Conflicts of Interest:** The authors declare no conflict of interest. The funders had no role in the design of the study; in the collection, analyses, or interpretation of data; in the writing of the manuscript, or in the decision to publish the results.

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
