# Peer review of "U-Net-Id, an Instance Segmentation Model for Building Extraction from Satellite Images—Case Study in the Joanópolis City, Brazil"

_remotesensing, doi:10.3390/rs12101544_

Round 1

Reviewer 1 Report

This is a very worthy piece of research, providing as it does the development of a new and viable method for deploying creative technology to assist in the mapping of the fast expanding towns and cities in Brazil. The development of such technology is critical for monitoring, planning and responding to the speed of urban development that is taking place in developing countries. You review the current technologies that exist for urban mapping, the rational and the approaches they deploy to gain the required information and honestly consider and review both the pros and cons of the available systems. The paper then presents and reports on a new approach – a U-based model for instance segmentation which you call U-net-id – which recognises the strengths of some of the current approaches to urban mapping – the Deep Watershed Transform for example, but describes in fine detail your new approach, the rational and detail behind this, the testing of the concept and the encouraging results. The text of necessity has a concentrated narrative, which requires a great deal of concentration on behalf of the reader to engage with the extensive information and discussion presented in the paper, but this is helped greatly by the quality of the figures deployed throughout the text, and the very useful text associated with each figure. Although suggestions for improving the research, potentially applying it across a much broader spectrum and the potential for carrying out further research is included in Section 4  Discussion, it might have been useful to expand Section 5 Conclusions a bit more, as this can provide a very useful back-up to the Abstract in communication the paper to a wider audience.

Author Response

FW : Following your recommendations, the conclusion has been expanded to give information to a wider audience, the paragraph reads now :

 « In this work, we present a new architecture of deep learning called U-net-id, specifically designed for the instance segmentation of buildings in very high resolution satellite imagery. The architecture is built to work with image in tiles, such as remote sensing images, and to reduce the at the minimum the information need to be provided by the user: only the images for prediction or only the images and the masks for training. No manual threshold has to be defined by the user, which is not the case in the currently most used instance segmentation frameworks such as Mask R-CNN or Deep Watershed Transform. The U-net-id architecture achieves great performance for instance segmentation of the instance building datasets of the city of Joanópolis-SP, Brazil. It will be further improved with samples from other Brazilian cities and applied to building instance segmentation of the main cities of Brazil. The Joanópolis dataset do not contain all the building morphologies that can be encountered in Brazil or in other countries, however to adapt the model to learn new features of buildings, the user will only need to provide additional training sample, the architecture will remain unchanged. Due to the model architecture simplicity, it could be easily reproduced and used for buildings or tested for instance segmentation of other objects in very high resolution images. The model R code is available at https://doi.org/10.5281/zenodo.3716070» 

Reviewer 2 Report

The paper is in general well written with some mistakes that should be corrected. 

The authors present a novel method for instance segmentation using 3 masks in a CNN. The datasets and procedure are detailed in their explanations as well as the results. It is recommended that not only the proposed method is tested with the sample dataset but a comparison be made even is the training masks differ for the methods.

The comparison would aid the conclusions that so far are lacking depth and do not represent the paper and its findings.

Author Response

FW : Dear Reviewer #2, thank you very much for you positive review. In the following text you will find your comments and our answers in blue.

Comments and Suggestions for Authors

The paper is in general well written with some mistakes that should be corrected.

The authors present a novel method for instance segmentation using 3 masks in a CNN. The datasets and procedure are detailed in their explanations as well as the results. It is recommended that not only the proposed method is tested with the sample dataset but a comparison be made even is the training masks differ for the methods.

The comparison would aid the conclusions that so far are lacking depth and do not represent the paper and its findings.

FW : Thank you for your comments, the Reviewer #3 has also commented this point.

Our model compares with Mask R-CNN (the most commonly used model for instance segmentation) or Deep Watershed Transform as fully detailed in the introduction L38 to L69 and in the discussion L248, to L282. There is no comparison with the results of other models because previously the cited models are not specifically designed for satellite imagery dataset and some adaptions should be made that will take more than weeks, if they can be resolved. Let me explain, in Mask R-CNN, some parameters are subjective, for example, one of the parameters to give is the number of objects (or anchor) to be found in the images. How to define this number? there is no objective answer to this question, that can be likely be only resolved by tests/errors. Our model does not suffer of such subjective parametrization (a previous probability of object presence) and segments objects as a CNN, that is, it processes every pixel of the image. We have the sentence in discussion L263 to clarify this : “ Furthermore, for Mask R-CNN, the number of objects to be found in the images must be given manually as a parameter (anchors) [6], and the results can differ based on this parameter. Our method does not have this limitation. It does not depend of a previous probability of object presence in the image, apply the same process to every pixel and is not sensitive to the number of buildings in the image.”

Furthermore, Mask R-CNN is designed for object segmentation in image, but not in tiled images such as in satellite images. In our model, every mask is coded in binary, 0 or 1. In the inner mask, two pixels adjacent and with the value 1 are of the same instance. This remain true when joining two tiles with an object overlapping the both tiles. The inner mask of the object in the image 1 and the image 2 is coded with 1 and will naturally give only one object when joining the two adjacent images. In Mask R-CNN, it is not like this, as each object have a unique id in each image. For the example, if an object overlap between two images, the two part of the object will have two different values (the id), how do the user decide to joint them or not ? here again, for Mask R-CNN there is no objective solution, only subjective ones. For Deep Watershed Transform, the problems are not so critical, as energy basins should be merged easily for object overlapping between two tiles, however, here again a manual defined threshold need to be set by the user to separate adjacent object on the value of the energy gradient. In our model, this is explicitly resolved in the inner segment masks where objects, even when adjacent, are separate by pixels of value equal zero.

The U-net-id model was designed to make an instance segmentation for tiled satellite images while resolving problems that can be encounter with the existing instance segmentation models, mainly Mask R-CNN and Deep Watershed Transform.

Following your comment and the comments or reviewer #1 and #3, we have developed and clarified the conclusion for a wider audience, the paragraph reads now : ” In this work, we present a new architecture of deep learning called U-net-id, specifically designed for the instance segmentation of buildings in very high resolution satellite imagery. The architecture is built to work with image in tiles, such as remote sensing images, and to reduce the at the minimum the information need to be provided by the user: only the images for prediction or only the images and the masks for training. No manual threshold has to be defined by the user, which is not the case in the currently most used instance segmentation frameworks such as Mask R-CNN or Deep Watershed Transform. The U-net-id architecture achieves great performance for instance segmentation of the instance building datasets of the city of Joanópolis-SP, Brazil. It will be further improved with samples from other Brazilian cities and applied to building instance segmentation of the main cities of Brazil. The Joanópolis dataset do not contain all the building morphologies that can be encountered in Brazil or in other countries, however to adapt the model to learn new features of buildings, the user will only need to provide additional training sample, the architecture will remain unchanged. Due to the model architecture simplicity, it could be easily reproduced and used for buildings or tested for instance segmentation of other objects in very high resolution images. The model R code is available at https://doi.org/10.5281/zenodo.3716070”

Reviewer 3 Report

In general, authors are presenting an interesting application of satellite image extraction using an instance segmentation model in form of letter (publication type).

Graphical results are shown in the form of comparative application images, in addition, corresponding numerical results are entirely merged within the text "and somehow confusing for both!". In this way, a compilation in a recapitulative table could be of great interest and clearness (as a simple recommendation).

At the technical level, are the training samples sufficient so that the model can be generalized for applications to whatever location and buildings distribution and morphology? If so, what are the model limitations (that are not clear in the letter!). In this case, authors are highly invited to add in their letter title the specific application locations ("and therefore the implicit imitations").

A second reflection; what about a comparative results with other segmentation models for your specific case? What are your arguments?

- The letter is clearly presenting results for a specific segmentation model for specific locations, for instance, the title is confusing and must not be generalized (authors must include: application case...).

- References part must absolutely be enriched and updated (2019 as recent one). In the bibliography, there are primordial segmentation papers (classical and very recent ones 2020) that are recommended for your specific case (to be included).

Author Response

FW: Dear Reviewer #3, thank you very much for your detailed revisions. We have answered at your comments and suggestions in detail, please find you answer in blue in the attached pdf.

Round 2

Reviewer 3 Report

Graphical results are shown in the form of comparative application images, in addition, corresponding numerical results are entirely merged within the text "and somehow confusing for both!". In this way, a compilation in a recapitulative table could be of great interest and clearness (as a simple recommendation).

>> Recommendation not taken into consideration.

At the technical level, are the training samples sufficient so that the model can be generalized for applications to whatever location and buildings distribution and morphology? If so, what are the model limitations (that are not clear in the letter!). In this case, authors are highly invited to add in their letter title the specific application locations ("and therefore the implicit imitations").

>> Title improvement (to be specific) is not taken into consideration.

>> About the model limitations; considerations not taken adequately into account (short but not sufficient paragraphs were updated to the discussion/results parts). The arguments must be added in the earlier beginning parts of the paper to expose either the advantages and the limitations of the proposed model.

A second reflection; what about a comparative results with other segmentation models for your specific case? What are your arguments?

>> idem (last remark).

The letter is clearly presenting results for a specific segmentation model for specific locations, for instance, the title is confusing and must not be generalized (authors must include: application case...).

>> Title improvement (to be specific) is not taken into consideration.

References part must absolutely be enriched and updated (2019 as recent one). In the bibliography, there are primordial segmentation papers (classical and very recent ones 2020) that are recommended for your specific case (to be included).

>> For the betterment of the letter the references part must absolutely be considerably updated with consistent papers.

Author Response

Please find our responses to your comment and suggestion in the attached pdf.
